# Mining the Mesenchymal Stromal Cell Secretome in Patients with Chronic Left Ventricular Dysfunction

**DOI:** 10.3390/cells11132092

**Published:** 2022-06-30

**Authors:** Jacquelynn Morrissey, Fernanda C. P. Mesquita, Lourdes Chacon-Alberty, Camila Hochman-Mendez

**Affiliations:** Department of Regenerative Medicine Research, Texas Heart Institute, Houston, TX 77030, USA; jacquelynn.morrissey@gmail.com (J.M.); fmesquita@texasheart.org (F.C.P.M.); lchacon@texasheart.org (L.C.-A.)

**Keywords:** clinical trials, FOCUS-CCTRN, heart failure, bedside-to-bench, mesenchymal stromal cells, secretome

## Abstract

Close examination of the initial results of cardiovascular cell therapy clinical trials indicates the importance of patient-specific differences on outcomes and the need to optimize or customize cell therapies. The fields of regenerative medicine and cell therapy have transitioned from using heterogeneous bone marrow mononuclear cells (BMMNCs) to mesenchymal stromal cells (MSCs), which are believed to elicit benefits through paracrine activity. Here, we examined MSCs from the BMMNCs of heart failure patients enrolled in the FOCUS-CCTRN trial. We sought to identify differences in MSCs between patients who improved and those who declined in heart function, regardless of treatment received. Although we did not observe differences in the cell profile of MSCs between groups, we did find significant differences in the MSC secretome profile between patients who improved or declined. We conclude that “mining” the MSC secretome may provide clues to better understand the impact of patient characteristics on outcomes after cell therapy and this knowledge can inform future cell therapy trials.

## Introduction

Over the last three decades, cell therapy has become a mainstay of regenerative medicine research. Positive treatment results in small and large animal models of myocardial infarction and heart failure generated excitement about the ability of autologous cells to repair the heart [1,2]. The safety of cell therapy has been shown in several clinical trials; however, the effectiveness of this therapy has been inconsistent [3]. In studying the effect of cell therapy on outcomes, researchers have examined cell preparation methods, cell potency and mechanisms [4], and patient characteristics [5,6]. Samples from cell therapy clinical trials stored in biorepository programs enable “bedside-to-bench” studies, which can lead to mechanistic insights and more effective cell-based treatments. As the limitations of heterogeneous bone marrow mononuclear cell (BMMNC) populations used in earlier cardiovascular clinical trials are now better understood, mesenchymal stromal cells (MSCs) are currently the leading candidate for cell therapy [7]. Here, we examined MSCs from BMMNCs of patients enrolled in the FOCUS-CCTRN trial to identify any differences in cells between individuals who improved or declined clinically in the trial, regardless of treatment received.

In most cell therapy trials for heart failure, endpoint outcome evaluations have shown limited to no benefit of cell treatment. However, access to patients’ baseline cell populations stored in biorepositories has opened new avenues of investigation into the effect of intrinsic patient characteristics on the response to therapy. One such embedded cohort study was conducted on cells from patients in the FOCUS-CCTRN trial to identify intrinsic differences between patients who improved in three cardiac function indicators (increase in left ventricular ejection fraction [LVEF] and maximal oxygen consumption [VO2max] and decrease in left ventricular end systolic volume [LVESV] at day 180 compared to the baseline values; *n* = 17, improvers) and those who did not improve (decrease in LVEF and VO2max and increase in LVESV at day 180 compared to the baseline values; *n* = 11, non-improvers) (Table 1), regardless of treatment received [5]. The FOCUS-CCTRN trial was a first-generation, multicenter, randomized trial that examined the use of autologous BMMNCs in patients with chronic ischemic heart failure (Table 2) [8]. Patients who demonstrated improvements in all three functional criteria listed above were deemed improvers, whereas those who exhibited decline in all three variables were considered non-improvers. The results showed that improvers had a higher expression of B-cell and CXCR4+ BMNNCs at baseline (Table 1). Furthermore, improvers also had fewer endothelial colony-forming cells, as quantified by endothelial colony-forming cell and CFU-Hill in their bone marrow as compared to non-improvers. Findings from this cohort analysis suggested that intrinsic patient characteristics may be critical to deriving clinical benefit from cell therapy [5].

Using whole BMMNCs to treat chronic ischemic heart disease has shown limited to mixed results [10,11], but isolating specific fractions of BMMNCs to enrich cell products for therapeutic use may be a cost-efficient and relatively simple procedure to improve outcomes [12]. BMMNCs comprise diverse cell populations, including immune cells, monocytes, and various progenitor cells, of which MSCs constitute up to 0.02% [13,14]. MSCs are also part of the resident stromal cell populations in various tissues [15], and despite comprising only a small portion of BMMNCs, they contribute significantly to cardiovascular repair by secreting various growth factors and protective molecules into the local microenvironment [14]. Considering the therapeutic potential of MSCs and knowing that clinical improvers and non-improvers are inherently different at baseline, we sought to characterize any differences in MSCs from patients who had different clinical outcomes in the FOCUS-CCTRN trial. We selected 3 of the 17 patients from the improver cohort and 3 of the 11 from the non-improver group who had BMMNCs cryopreserved in our biorepository. Subjects were matched by age and comorbidities (Table 3).

Although BMMNCs from patients in the FOCUS-CCTRN trial had been cryopreserved in biorepositories for nearly a decade, we were able to thaw the samples and isolate and expand the MSCs in vitro for up to 10 passages (*n* = 3 improvers and *n* = 3 non-improvers). MSCs from both clinical improvers and non-improvers retained the characteristic immunophenotype as determined by flow cytometry, and their ability to direct differentiation of patient MSCs into adipocytes, osteocytes, and chondrocytes was not impaired by clinical status (Figure 1A, C, and D). Because cell senescence is known to inhibit cell plasticity [16], we evaluated the telomere length of improver and non-improver MSCs at passage 5; no significant difference was observed between groups (Figure 1B). Notably, MSCs derived from FOCUS-CCTRN patients showed no phenotypic or characteristic differences from MSCs of healthy donors [17,18].

The potential benefits of MSCs in tissue repair are attributed to paracrine actions resulting from the biologically active molecules secreted in response to the local environment. This was confirmed when conditioned medium from MSCs was shown to repair and protect damaged myocardium in infarct models [12]. MSCs’ diverse secretome includes growth factors, nucleic acids, and exosomes, which are released into the injured microenvironment [19]. In recent years, MSC-derived exosomes (EXO-MSC), specifically their microRNA cargo, have been extensively examined in MSC-based cell therapy for heart repair [20]. Preclinical data suggest that exosomal microRNA contributes to cardioprotection [21].

In the setting of cardiac infarct, microRNA-containing EXO-MSCs regulate immune cell activation and inflammation [22] while also mitigating the effects of hypoxia-induced apoptosis [23]. Specifically, miR-4732-3p, which was upregulated among the improver cohort in this study, exerts protection through its effects on various cardiac cell types, limiting fibrosis, enhancing angiogenesis, and insulating cardiomyocytes against reactive oxygen species and cell death [24].

To examine if EXO-MSC content was associated with the positive outcomes in the improver cohort, we performed microRNA sequencing of EXO isolated from cell supernatant of both improvers (*n* = 3) and non-improvers (*n* = 3). After preparing the libraries, we pooled the microRNA and sequenced it using the NextSeq 500 (Illumina). Using edgeR, we identified 16 upregulated and 12 downregulated microRNAs in improvers versus non-improvers (Table 4). Additionally, we identified 1968 target genes associated with overexpressed microRNA and enrichment analysis using the Reactome database, which showed 74 statistically significant pathways (adjusted *p* value < 0.05); these pathways included the VEGFA-VEGFR2 pathway (yellow dots), cellular responses to stress (green dots), programmed cell death (red dots), and signaling by SCF-KIT (blue dots) (Figure 2A). When analyzing the under-expressed microRNA (Figure 2B), 976 target genes were associated with these microRNAs and the enrichment analysis showed 100 statistically significant pathways, including downregulation of the immune system (yellow dots), TGF-beta receptor signaling (green dots), and VEGFR2-mediated regulation (red dots). Since we normalized the data for possible confounders, we assumed that the differences observed in EXO-MSCs were not related to patient demographics (Table 3), suggesting an intrinsic differential effect of EXO between the groups. Further study into the significance of these pathways in patients with chronic ischemic heart disease is necessary to understand the beneficial effects of MSC paracrine activity.

Reevaluation of the data from first-generation clinical trials suggests the importance of differences in baseline patient characteristics. Updates to refine the phenotypic characterization of MSCs [25] and enhance cell product uniformity have been made in recent years. Nonetheless, the traditional release criteria for the clinical use of MSCs hinge on the 2006 International Society of Stem Cell Research (ISSCR) standards related to immunophenotypic markers, plastic adherence, in addition to tripotential differentiation into the adipogenic, osteogenic, and chondrogenic lineages [26]. However, our data indicate that traditional cell characterization using these ISSCR standards falls short of indicating factors that may prove useful in discerning patient outcomes. More importantly, our results suggest that the EXO-MSC of improvers in the FOCUS-CCTRN trial contained microRNAs related to various signaling pathways and cell structure components that were lacking in non-improver patients. Thus, the MSC secretome may provide important clues to help understand the mechanisms of cardiac repair in the setting of ischemic heart disease and may help to explain why patients experience different clinical outcomes. Furthermore, the secretome may be a relevant parameter to include in MSC release criteria, substantiating current efforts to alleviate inconsistencies regarding the impact of cell therapy, which may be attributable to non-rigorous cell sourcing/definition or donor characteristics [25,27,28].

Although early cardiovascular clinical trials did not consistently show benefit from cell therapy, examining samples from biorepositories via emerging techniques such as microRNA sequencing can help identify key differences between patient cell cargo that may provide insight for future clinical trial design. Combining the knowledge of basic research with detailed clinical features of patients can help to deliver the promise of personalized medicine in cell therapy. 

## Figures and Tables

**Figure 1 cells-11-02092-f001:**
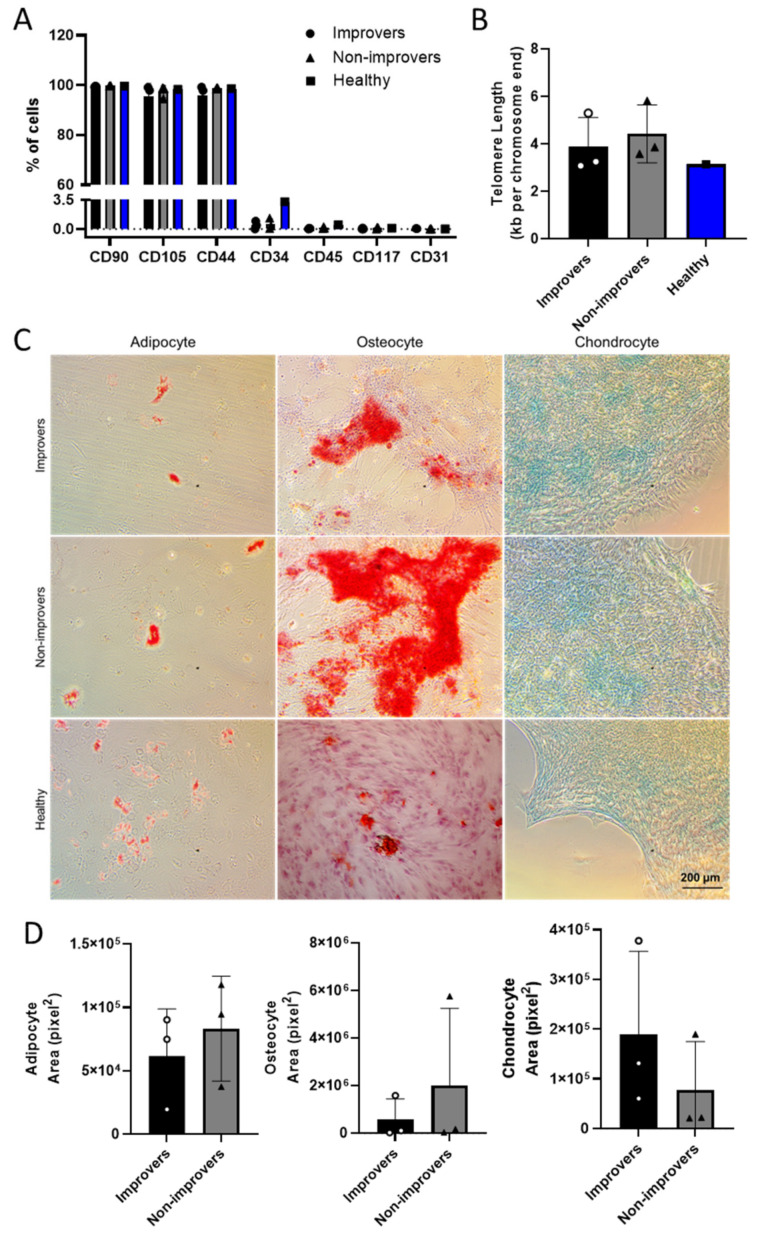
Characterization of MSCs derived from BMMNCs of improvers and non-improvers in passage 7 (*n* = 3). (**A**) Expression of surface molecules by flow cytometry (*n* = 3, improvers; *n* = 3, non-improvers; *n* = 1, healthy donor). (**B**) Telomere length of MSCs. (**C**) Representative images of differentiation capacity of MSC. Adipogenic differentiation showing lipid vacuoles in red (left). Osteogenic differentiation showing calcium deposits in red (middle). Chondrogenic differentiation showing glycosaminoglycans in blue (right). (**D**) Quantification of trilineage potential of MSC (*n* = 3).

**Figure 2 cells-11-02092-f002:**
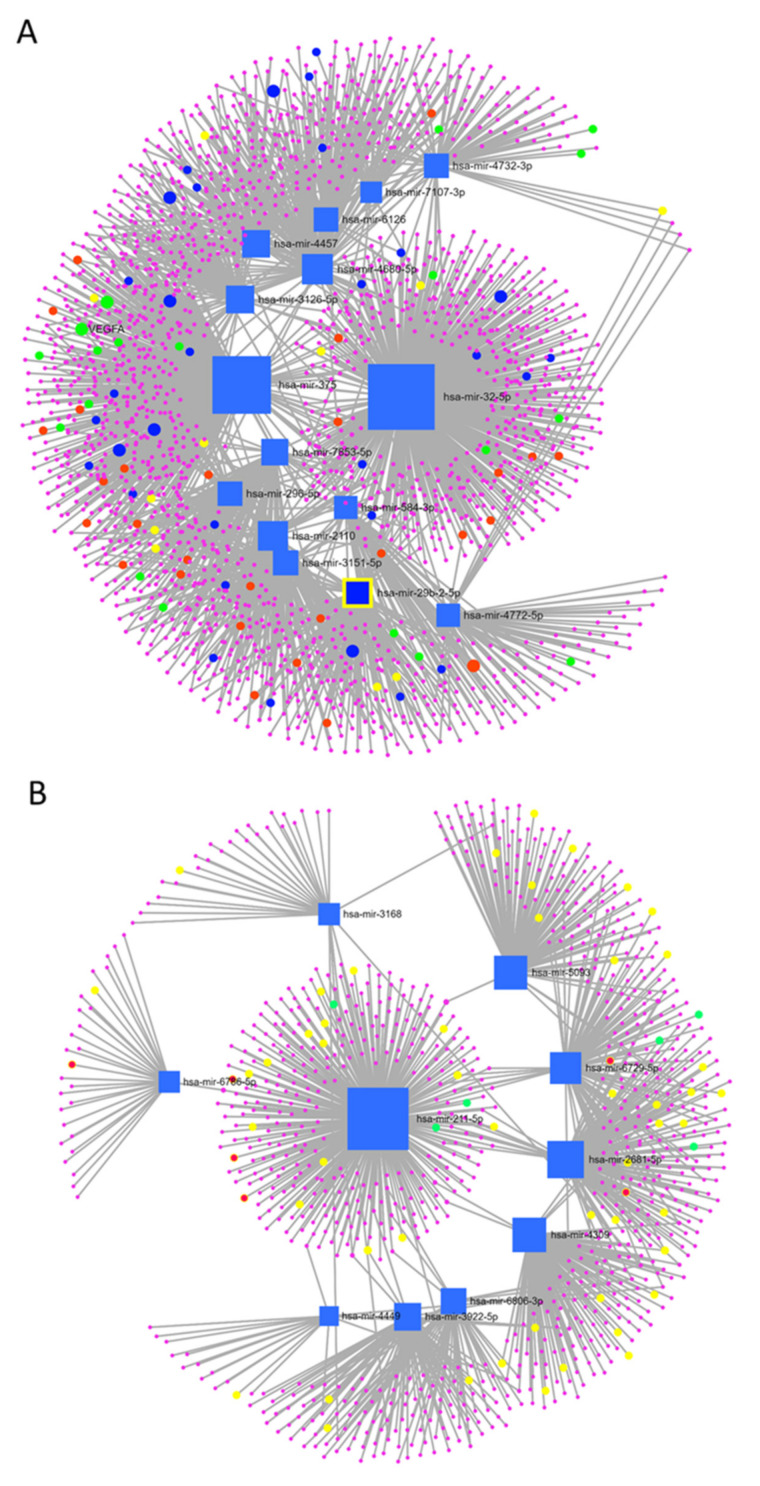
Network analysis showing the genes associated with differentially expressed microRNA (*n* = 3). (A) Network analysis showing the genes associated with the overexpressed miRNA and enrichment analysis. (B) Network analysis showing the genes associated with the under-expressed microRNA: immune system and enrichment analysis (yellow dots), downregulation of TGF-beta receptor signaling (green dots), and VEGFR2-mediated regulation (red dots) when comparing improver and non-improver cohorts.

**Table 1 cells-11-02092-t001:** FOCUS-CCTRN embedded cohort analysis: Bedside-to-bench outcomes highlight patient subpopulations with distinct BMMNC profiles.

Reference	Clinical Improvers (*n* = 17)	Improver BMMNC Characteristics
[5]	Decreased LVESV, increased VO2 max, and increased LVEF	Higher CD19+ B cells, CD11b+ monocytes, CD31dim cells, and CXCR4+ cells; lower CD31bright cells

BMMNC, bone marrow mononuclear cell; LVEF, left ventricular ejection fraction; LVESV, left ventricular end systolic volume.

**Table 2 cells-11-02092-t002:** FOCUS-CCTRN trial: Patient demographics, study design, and clinical criteria.

Reference	Category	Characteristic
[8,9]	Patient demographics (enrollment)	Age: ≥18 years old with CAD, LVEF ≤ 45% and limiting angina and/or symptomatic CHF
	Study size	153 patients consented; 92 randomized to receive treatment, 6 excluded from treatment
	Study design	Randomized 2:1 for treatment (100 × 10^6^ BMMNCs) vs. control (5% human albumin)
	BMMNC isolation technique	Sepax (Ficoll method)
	Release criteria	CD34^+^/CD133^+^; CFU-GM colony growth
	Product delivery	Within 12 h of bone marrow aspiration
	Follow-up	6 months after delivery
	Primary endpoints	Left ventricular end-systolic volume, maximal oxygen consumption, infarct size
	Exploratory outcomes	Increased left ventricular ejection fraction

**Table 3 cells-11-02092-t003:** Patient and BMMNC characteristics from samples selected for MSC isolation.

Category	Characteristics	Improvers (*n* = 3)	Non-Improvers (*n* = 3)
Patient demographics and primary endpoints	Age	60.03 ± 5.92	67.71 ± 7.96
	Diabetes	No	No
	Hypertension	Yes	Yes
	Hyperlipidemia	Yes	Yes
	Smoking	Yes	Yes
	LVEF at baseline (%)	35.17 ± 13.2	38.3 ± 7.52
	LVEF at endpoint (%)	37.67 ± 14.32	32.73 ± 8.68
	LVESV at baseline (%)	103.1 ± 32.1	154.4 ± 58.11
	LVESV at endpoint (%)	128.4 ± 46.32	137.4 ± 61.58
BMMNC characteristics (at baseline)	Viability (%)	99 ± 1	97.67 ± 0.57
	CD34/CD133 (Mean %, *p* = 0.49)	2.29	1.48
	CD19 (Mean %, *p* = 0.79)	11.51	13.11
	CD11b (Mean %, *p* = 0.98)	62.62	62.43
	CXCR4 (Mean %, *p* = 0.86)	63.12	61.9
	CD31^dim^ (Mean %, *p* = 0.75)	9.36	8.57
	CD31^bright^ (Mean %, *p* = 0.38)	0.12	0.25

BMMNC, bone marrow mononuclear cell; LVEF, left ventricular ejection fraction; LVESV, left ventricular end systolic volume.

**Table 4 cells-11-02092-t004:** Mining MSC-derived exosome sequencing: Differential expression of MSC cargo between clinical improvers and non-improvers.

Over-Expressed Improver miRNAs		Under-Expressed Improver miRNAs	
hsa-miR-4680-5p	hsa-miR-584-3p	hsa-miR-2681-5p	hsa-miR-211-5p
hsa-miR-375	hsa-miR-7107-3p	hsa-miR-5093	hsa-miR-6729-5p
hsa-miR-32-5p	hsa-miR-296-5p	hsa-miR-3168	hsa-miR-4449
hsa-miR-4457	hsa-miR-6126	hsa-miR-6806-3p	hsa-miR-3922-5p
hsa-miR-3126-5p	hsa-miR-3151-5p	hsa-miR-4309	hsa-miR-6786-5p
hsa-miR-2110	hsa-miR-4732-3p	has-miR-019914	has-miR-004153
hsa-miR-29b-2-5p	hsa-miR-7853-5p		
hsa-miR-105-3p	hsa-miR-4772-5p		

## Data Availability

The data presented in this study are available on request from the corresponding author.

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
