# Peer review of "Mining the Mesenchymal Stromal Cell Secretome in Patients with Chronic Left Ventricular Dysfunction"

_cells, 2022, doi:10.3390/cells11132092_

Round 1

Reviewer 1 Report

Please mention how many improvers x non-improvers you have in the trial, and how many patients have you evaluated for each outcome you show afterwards.

Do the authors believe that with the evaluated number of patients, the study has enough power to conclude the analysis?

Page 2: Please define/clarify what do you mean by: 1- incrased LVESV (how much increase? or are you coparing to values in baseline?). 2- Increased VO2 (how much increase? or are you comparing to values in baseline?). 3 - Increased LVEF (how much increase? or are you coparing to values in baseline?)

There is conflicting information in the text (line 54)and in the table 1 regarding to monocytes/macrophages: Improvers have fewer or higher amount of CD11b+ cells?

Page 3 - Osteocyte differentiation looks quite different from improvers then non-improvers. Please change the pictures, or if there is difference, please comment on that in the text.

Page 4 - Line 98 - Chemoattractants cause MSC to travel from several places to sites of injury, not just from bone marrow. Please correct it. 

Page 5: Are the authors suggesting that secretome be used as a relevant parameter to MSC definition or for MSC use in clincal trials?

Author Response

General comment from authors

Thank you for your comments and suggestions. The samples used in this opinion were collected about 10 years ago in a clinical trial that used bone marrow-mononuclear cells (BMMNCs) as part of the therapy. The biorepository received and stored only BMMNCs that were not used for the treatment procedure, thus resulting in a limited and varied amount and number of stored patients’ products. Additionally, the stored samples were used in other ancillary studies, which made the sample pool even smaller. Because of the rarity of the specimens used, the authors chose the Opinion format, highlighting the findings and allowing future deeper investigation. However, the manuscript was erroneously labeled originally as a “Review” instead of an “Opinion” piece, leading to enormous confusion. We have reformatted the text–using the correct template–to reflect the proper category.

REVIEWER 1:

Comments and Suggestions for Authors

Please mention how many improvers x non-improvers you have in the trial, and how many patients have you evaluated for each outcome you show afterwards.

Response: We appreciate the reviewer’s comment and want to clarify this issue. The FOCUS-CCTRN trial had 17 improvers and 11 non-improvers. For this study, we selected 3 from each cohort. To clarify this point, we added the N values of each cohort in lines 48 and 50 of the text and in Tables 1 and 3. Additionally, we clarified this point in lines 79-80, as shown below.

We selected 3 of the 17 patients from the improver cohort, and 3 of the 11 from the non-improver group who had BMMNCs cryopreserved in our biorepository.

Do the authors believe that with the evaluated number of patients, the study has enough power to conclude the analysis?

Response: The number of patients used in the original study to identify improvers versus non-improvers was 78 (see citation below); 17 patients were demonstrated to be improvers, whereas 11 declined in all three specific functional endpoints (described as non-improvers). The authors demonstrated statistical differences between the groups. For this study, we selected samples of matching age and comorbidities (Table 3) to minimize confounders, which resulted in having 3 samples per group. Because of the low sample size, we chose to submit our study as an Opinion piece rather than an Original Investigation; however, we believe our findings, even with a small sample size,  provide relevant and important insights for the fields of cell therapy and regenerative medicine.

Taylor D.A., Perin E.C., Willerson J.T., Zierold C., Resende M., Carlson M., Nestor B., Wise E., Orozco A., Pepine C.J., Henry T.D., Ellis S.G., Zhao D.X., Traverse J.H., Cooke J.P., Schutt R.C., Bhatnagar A., Grant M.B., Lai D., Johnstone B.H., Sayre S.L., Moye L., Ebert R.F., Bolli R., Simari R.D., Cogle C.R., Cardiovascular Cell Therapy Research N. Identification of Bone Marrow Cell Subpopulations Associated With Improved Functional Outcomes in Patients With Chronic Left Ventricular Dysfunction: An Embedded Cohort Evaluation of the FOCUS-CCTRN Trial. Cell Transplant. 2016,25,1675-87.

Page 2: Please define/clarify what do you mean by: 1- increased LVESV (how much increase? or are you comparing to values in baseline?). 2- Increased VO2 (how much increase? or are you comparing to values in baseline?). 3 - Increased LVEF (how much increase? or are you comparing to values in baseline?)

Response:  We thank the reviewer for this comment and apologize for any confusion. The increases or decreases in the above-mentioned variables were indeed compared to baseline values. In addition, there was no set threshold for the amount of increase or decrease identified. We have clarified the patient grouping criteria in the manuscript (lines 45-50) as shown below.

(…) intrinsic differences between patients who improved in three available cardiac function indicators (increase in left ventricular ejection fraction [LVEF] and maximal oxygen consumption [VO2max] and decrease in left ventricular end systolic volume [LVESV] at day 180 compared to the baseline values; n=17, improvers) and those who did not improve (decrease in LVEF and VO2max and increase in LVESV at day 180 compared to the baseline values; n=11, non-improvers) (Table 1), regardless of treatment received [5].

There is conflicting information in the text (line 54) and in the table 1 regarding to monocytes/macrophages: Improvers have fewer or higher amount of CD11b+ cells?

Response: We apologize for this oversight. We have now updated the text to reflect the higher expression of CD11b+ monocytes in the improver population.

Page 3 - Osteocyte differentiation looks quite different from improvers then non-improvers. Please change the pictures, or if there is difference, please comment on that in the text.

Response: We have now updated the image to reflect the similarity in the osteocyte differentiation of improvers and non-improvers (see new image for Figure 1C).

Page 4 - Line 98 - Chemoattractants cause MSC to travel from several places to sites of injury, not just from bone marrow. Please correct it. 

Response: To avoid misinterpretation, we have removed the sentence and have now updated the text to focus on cardioprotection and cardiac repair.

Page 5: Are the authors suggesting that secretome be used as a relevant parameter to MSC definition or for MSC use in clincal trials?

Response: In most cell therapy trials for heart failure, endpoint evaluations have shown limited to no benefit of cell treatment. The re-evaluation of the data from these clinical trials suggests underlying differences at the baseline level of the “cell donor,” regardless of the treatment received during the trial. The paracrine hypothesis of cell repair is now the most widely accepted mechanism for cell therapy, and exosomes have emerged as a promising reparative candidate. Because of this, we believe that the secretome content needs to be included as a variable for evaluating and accepting cell-derived products, which would improve the rigor of the cell product and limit donor-specific variability.

Reviewer 2 Report

The authors examined decade-old MSCs obtained from BMMNCs of heart failure patients previously enrolled in the FOCUS-CCTRN trial. They try to find relevant differences in MSCs between patients who improved and those who declined heart function regardless of treatment received. Although they did not find differences in MSCs profiles between improvers and non-improvers, the authors demonstrated that MSCs’ secretome display significant differences regarding the composition of microRNA released through Exosomes. The idea of the study is interesting, but the authors can improve the data presentation by providing more background information instead of only citing published references, as well as, exploring a bit deeper bioinformatic tools to present secretome results, as I will comment on below.

1 – Authors mentioned that MSCs were thawed and cultivated up to passage 10. What passages were used for each FACS, differentiations, telomere length, and secretome assays? These pieces of information were never clearly stated, and we do not have a methods section as a reference. Add this information at least in the Figure legend. Also, what were the experimental sample sizes? Explain in terms of biological, experimental, and technical replicates.

2 – Telomere length is a proxy for cell senescence (cell aging). There is a list of standard articles published demonstrating MSCs aging as a continuous effect of long-term cultivation in vitro. The authors compared improvers and non-improvers, but no healthy control cells were shown. Once improvers display significant differences in exosome contents, a better comparison of these subjects with healthy (non-HF individuals) may elucidate critical pathways differentially expressed in both improvers and healthy individuals. Elaborate a bit more on this discussion, please.

3 – Differentiation assays for adipogenesis, osteogenesis, and chondrogenesis are standard in the field. Usually, the results presented for these assays are more convincing than those printed in this manuscript. I’ve been working with these assays for the last decades, and I only experienced low efficiency like that using pig cells. Why do you think your results are so weak? Did you try them in lower passages? How many patients were tested for each of the diffs. protocols? Do you believe that the “enhanced osteogenesis” observed in non-improvers can partially explain why these patients had the worst prognosis? Elaborate, please?

4 – If you have data for each differentiation protocol, please try to create a metric system to quantify the efficiency of each protocol and compare groups.

5 – Exosomes were never properly introduced. Paracrine effects are not only due to exosomes. Please elaborate better this transition from cells to released “goods” and finally emphasize exosomes. How did you isolate and quantified exosomes characteristics? Authors never mentioned methods for isolation and characterization.

6 – Authors never explained in totality their sampling and exosome contents analysis. Why did you decided to access only microRNAs? What was the sequencing methodology? What was the sequencing analysis pipeline?

7 – Please, explore sequencing analysis, microRNA Reactome, and other bioinformatic relevant tools to represent your findings better. Many open-source tools are available in the literature to examine gene/miRNA interactions. These results are pretty interesting to be only mentioned as a textual paragraph. The authors should elaborate more on this comparison. Secretome analysis is the “cherry” here.

Author Response

General comment from authors

Thank you for your comments and suggestions. The samples used in this opinion were collected about 10 years ago in a clinical trial that used bone marrow-mononuclear cells (BMMNCs) as part of the therapy. The biorepository received and stored only BMMNCs that were not used for the treatment procedure, thus resulting in a limited and varied amount and number of stored patients’ products. Additionally, the stored samples were used in other ancillary studies, which made the sample pool even smaller. Because of the rarity of the specimens used, the authors chose the Opinion format, highlighting the findings and allowing future deeper investigation. However, the manuscript was erroneously labeled originally as a “Review” instead of an “Opinion” piece, leading to enormous confusion. We have reformatted the text–using the correct template–to reflect the proper category.

REVIEWER 2:

Comments and Suggestions for Authors

The authors examined decade-old MSCs obtained from BMMNCs of heart failure patients previously enrolled in the FOCUS-CCTRN trial. They try to find relevant differences in MSCs between patients who improved and those who declined heart function regardless of treatment received. Although they did not find differences in MSCs profiles between improvers and non-improvers, the authors demonstrated that MSCs’ secretome display significant differences regarding the composition of microRNA released through Exosomes. The idea of the study is interesting, but the authors can improve the data presentation by providing more background information instead of only citing published references, as well as, exploring a bit deeper bioinformatic tools to present secretome results, as I will comment on below.

1 – Authors mentioned that MSCs were thawed and cultivated up to passage 10. What passages were used for each FACS, differentiations, telomere length, and secretome assays? These pieces of information were never clearly stated, and we do not have a methods section as a reference. Add this information at least in the Figure legend. Also, what were the experimental sample sizes? Explain in terms of biological, experimental, and technical replicates.

Response: We thank the reviewer for these important comments. As per the suggestion, we have now updated the text to address these issues. We have clarified the number of samples and the cell passages used in our experiments throughout the text and in the figure legends. We used BMMNCs from 3 improvers and 3 non-improvers to isolate the MSCs. The data shown are biological replicates of each group. In flow cytometry and telomere length analyses, we used MSCs derived from a healthy donor (commercially available) as a control.

2 – Telomere length is a proxy for cell senescence (cell aging). There is a list of standard articles published demonstrating MSCs aging as a continuous effect of long-term cultivation in vitro. The authors compared improvers and non-improvers, but no healthy control cells were shown. Once improvers display significant differences in exosome contents, a better comparison of these subjects with healthy (non-HF individuals) may elucidate critical pathways differentially expressed in both improvers and healthy individuals. Elaborate a bit more on this discussion, please. –

Response: We evaluated telomere length of MSCs by using the Absolute Human Telomere Length Quantification qPCR Assay Kit. Knowing that the long-term cultivation of MSC is a parameter for cell senescence, we performed the quantification on cells in the same passage (passage 7). In addition, we did include MSCs from a healthy donor as controls, also in passage 7 to reflect the same cell age. Our results indicated no numerical difference in any groups with MSCs of the same cell age/passage: improvers, 3.71±1.06 (n=3); non-improvers, 4.25±1.06 (n=3); and healthy donors, 3.14 (n=1). We have now included the data on telomere length in healthy donor MSCs in the bar graph in Figure 1B, as per the reviewer’s suggestion.

3 – Differentiation assays for adipogenesis, osteogenesis, and chondrogenesis are standard in the field. Usually, the results presented for these assays are more convincing than those printed in this manuscript. I’ve been working with these assays for the last decades, and I only experienced low efficiency like that using pig cells. Why do you think your results are so weak? Did you try them in lower passages? How many patients were tested for each of the diffs. protocols? Do you believe that the “enhanced osteogenesis” observed in non-improvers can partially explain why these patients had the worst prognosis? Elaborate, please?

Response: To generate as much data possible from the limited samples, we expanded the MSCs to large passages (up to 10 passages), affecting the cells' differentiation efficiency. Control cells (derived from commercially available healthy subjects) were expanded similarly to mirror the patients’ cells for all studies to normalize the large passage effects. To clarify that there is no “enhanced osteogenesis,” we quantified the differentiation capacity, and our data indicated no differences between groups (please see the new Figure 1D)

4 – If you have data for each differentiation protocol, please try to create a metric system to quantify the efficiency of each protocol and compare groups.

Response: We have now added a new panel to Figure 1 showing the quantification of trilineage differentiation potency in improvers and non-improvers. Please see the new Figure 1D.

5 – Exosomes were never properly introduced. Paracrine effects are not only due to exosomes. Please elaborate better this transition from cells to released “goods” and finally emphasize exosomes. How did you isolate and quantified exosomes characteristics? Authors never mentioned methods for isolation and characterization.

Response: We thank  the reviewer for this helpful comment and agree that we need to properly introduce exosomes in the manuscript. In the paragraph on the paracrine actions of MSCs, we have now added a better description of exosomes and microRNA in transitioning the discussion from cells to secreted exosomes and in emphasizing the importance of exosome content and the microRNA cargo on it (see below and lines 119-122). In addition, we have added two new references to the manuscript to support the statement (references 20 and 21).

In recent years, MSC-derived exosomes (EXO-MSC), and specifically their microRNA car-go, have been extensively examined for their role in MSC-based cell therapy aimed at repairing the damaged heart, and exosomal microRNA was been shown to contribute to cardioprotection in preclinical studies (20,21).

As for the methodology for exosome isolation and extraction, we used the exoRNeasy kit from Qiagen. Per the reviewer’s comment, we have now added this methodology to the text and expanded that paragraph ( lines 119-122).

6 – Authors never explained in totality their sampling and exosome contents analysis. Why did you decided to access only microRNAs? What was the sequencing methodology? What was the sequencing analysis pipeline?

Response: As per the reviewer’s comment, we have included the sample size for the microRNA differential expression analysis (n=3 per group). The rationale for performing microRNA sequencing was based on literature reports showing the role of microRNA in cardiac repair. We have now better introduced this aspect of the study and have also added the sequencing methodology and pipeline (lines 119-137).

7 – Please, explore sequencing analysis, microRNA Reactome, and other bioinformatic relevant tools to represent your findings better. Many open-source tools are available in the literature to examine gene/miRNA interactions. These results are pretty interesting to be only mentioned as a textual paragraph. The authors should elaborate more on this comparison. Secretome analysis is the “cherry” here.

Response: We thank the reviewer for this comment and appreciate the positive interest. We identified target genes associated with the over-expressed and under-expressed microRNA. To specifically address the reviewer’s comment, we have now included these results in the new Figure 2 (lines 122-137). We also used the open-sourced Reactome database to perform a functional analysis; please see the new text (lines 138-156).

Reviewer 3 Report

In this concise review of the FOCUSCCTRN trial, the authors review specific patient characteristics that could contribute to the understanding of the mechanisms behind cell therapy improvers in heart failure. While the topic is interesting, the manuscript is a bit thin either for an article or review for publication in this journal.

Because the authors apply some analysis, it is recommended that this be published as an article, or at minimum, methods details be added. Specifically, for the miR differential expression analysis, what statistical model was used (improver vs non improver?)? Were cell type abundances (from flow) used as covariates, as the results may simply reflect cell abundance/proportions, especially considering how few samples there are. In addition, it'd be worth citing 3-5 previous studies with more discussion that has explored microRNA/exosomes or at least contextualizing the identified microRNA as being novel or consistent with previous literature. 

Author Response

General comment from authors

Thank you for your comments and suggestions. The samples used in this opinion were collected about 10 years ago in a clinical trial that used bone marrow-mononuclear cells (BMMNCs) as part of the therapy. The biorepository received and stored only BMMNCs that were not used for the treatment procedure, thus resulting in a limited and varied amount and number of stored patients’ products. Additionally, the stored samples were used in other ancillary studies, which made the sample pool even smaller. Because of the rarity of the specimens used, the authors chose the Opinion format, highlighting the findings and allowing future deeper investigation. However, the manuscript was erroneously labeled originally as a “Review” instead of an “Opinion” piece, leading to enormous confusion. We have reformatted the text–using the correct template–to reflect the proper category.

REVIEWER 3:

In this concise review of the FOCUSCCTRN trial, the authors review specific patient characteristics that could contribute to the understanding of the mechanisms behind cell therapy improvers in heart failure. While the topic is interesting, the manuscript is a bit thin either for an article or review for publication in this journal.

Because the authors apply some analysis, it is recommended that this be published as an article, or at minimum, methods details be added. Specifically, for the miR differential expression analysis, what statistical model was used (improver vs non improver?)? Were cell type abundances (from flow) used as covariates, as the results may simply reflect cell abundance/proportions, especially considering how few samples there are. In addition, it’d be worth citing 3-5 previous studies with more discussion that has explored microRNA/exosomes or at least contextualizing the identified microRNA as being novel or consistent with previous literature. 

Response: As we stated in our general comment, we chose the Opinion format to highlight our findings for this manuscript because of the unique situation of having a small, but rare pool of cell samples. However, we agree with the reviewer that adding method details would be helpful for the reader. Thus, we have now included information regarding exosome isolation and miR expression (expanding the bioinformatics analysis). In addition, we have now added more references (see below) to the discussion paragraph to reflect the reviewer’s suggestions.

  • Zhao, J.; Li, X.; Hu, J.; Chen, F.; Qiao, S.; Sun, X.; Gao, L.; Xie, J.; Xu, B. Mesenchymal stromal cell-derived exosomes attenuate myocardial ischaemia-reperfusion injury through miR-182-regulated macrophage polarization. Cardiovasc Res 2019 (reference 22)
  • Cheng, H.; Chang, S.; Xu, R.; Chen, L.; Song, X.; Wu, J.; Qian, J.; Zou, Y.; Ma, J. Hypoxia-challenged MSC-derived exosomes deliver miR-210 to attenuate post-infarction cardiac apoptosis. Stem Cell Res Ther 2020 (reference 23)
  • Sanchez-Sanchez, R.; Gomez-Ferrer, M.; Reinal, I.; Buigues, M.; Villanueva-Badenas, E.; Ontoria-Oviedo, I.; Hernandiz, A.; Gonzalez-King, H.; Peiro-Molina, E.; Dorronsoro, A.; et al. miR-4732-3p in Extracellular Vesicles From Mesenchymal Stromal Cells Is Cardioprotective During Myocardial Ischemia. Front Cell Dev Biol 2021 (reference 24)

Regarding cell type abundance, we have not attributed any outcome from the clinical trial to the MSC population in this study. So we did not perform any correction or use cell abundance as a covariate from our flow cytometry data.

Round 2

Reviewer 2 Report

Figure 1:

I'm still missing the control cells differentiation images in fig.1C. This is not mandatory at this point, but I'd like to see it in this Fig.

Figure 2:

I really appreciate that you explored a bit more Reactome in your data. Why don't you add some pathway enrichment analysis too? You can use free tools such as EnrichR. Although you mentioned the most relevant genes/pathways in the text, a more graphical and digested presentation can benefit your work.

For yes or no please comment.

Author Response

Reviewer 2:

Figure 1:

I'm still missing the control cells differentiation images in fig.1C. This is not mandatory at this point, but I'd like to see it in this Fig.

Response: We thank the reviewer for this comment. As per the suggestion, we have now included images of healthy donor MSCs in Figure 1C (page 4) to address the issue.

Figure 2:

I really appreciate that you explored a bit more Reactome in your data. Why don’t you add some pathway enrichment analysis too? You can use free tools such as EnrichR. Although you mentioned the most relevant genes/pathways in the text, a more graphical and digested presentation can benefit your work.

For yes or no please comment.

Response: Although we agree with the reviewer on the importance of enrichment-based analysis to identify functional changes and elucidate underlying functional mechanisms, we believe that an enrichment analysis could generate noise/bias in the current target gene prediction and pathway analysis due to the multiple statistical corrections needed to elucidate the interaction networks.

Reviewer 3 Report

The new revisions are appreciated and significantly help clarify/convey the authors' original intent. Upon further review, given the small sample sizes, please add dots to each bar graph where each data point is and/or change to box-and-whisker plots. After this change, the manuscript will be ready for publication.

Author Response

Reviewer 3:

The new revisions are appreciated and significantly help clarify/convey the authors' original intent. Upon further review, given the small sample sizes, please add dots to each bar graph where each data point is and/or change to box-and-whisker plots. After this change, the manuscript will be ready for publication.

Response: We thank the reviewer for the positive comments and the suggestion. We have now updated the graphics and have included the individual values to address the issue.